# Age and familiarity effects on musical memory

**Sarah A. Sauvé**[1,2]*, **Praveena Satkunarajah**[1], **Stephen Cooke**[1], **Özgen Demirkaplan**[1], **Alicia Follett**[1], **Benjamin Rich Zendel**[1]

1 Division of Community Health and Humanities, Faculty of Medicine, Memorial University of Newfoundland, St. John's, Newfoundland and Labrador, Canada, 2 School of Psychology, College of Health and Science, University of Lincoln, Lincoln, Lincolnshire, United Kingdom

* ssauve@lincoln.ac.uk

**Data Availability Statement:** All data and materials are available on the Open Science Framework (DOI 10.17605/OSF.IO/AY37P).

**Funding:** The author(s) disclosed receipt of the following financial support for the research,

## Abstract

### Background

A common complaint in older adults is trouble with their memory, especially for new information. Current knowledge about normal aging and changes in memory identify a divide between memory tasks that are unaffected by aging and those that are. Among the unaffected are recognition tasks. These memory tasks rely on accessing well-known information, often include environmental support, and tend to be automatic. Negative age effects on memory are often observed at both encoding and during recall. Older adults often have difficulty with recall tasks, particularly those that require effortful self-initiated processing, episodic memory, and retention of information about contextual cues. Research in memory for music in healthy aging suggests a skill-invariance hypothesis: that age effects dominate when general-purpose cognitive mechanisms are needed to perform the musical task at hand, while experience effects dominate when music-specific knowledge is needed to perform the task [1].

### Aims

The goals of this pair of studies were to investigate the effects of age and familiarity on musical memory in the context of real pieces of music, and to compare a live concert experimental setting with a lab-based experimental setting.

### Method

Participants' task was to click a button (or press the spacebar) when they heard the target theme in three pieces of music. One was Mozart's Eine Kleine Nachtmusik and the others were original pieces commissioned for this study, one tonal and one atonal. Participants heard the relevant theme three times before listening to a piece of music. The music was performed by the Newfoundland Symphony Orchestra; participants either attended the concert, or watched a recording of the concert in the lab. Participants also completed two short cognitive tests and filled out a questionnaire collecting demographic information and a hearing abilities self-assessment.

authorship, and/or publication of this article: this work was supported by BRZ's Natural Sciences and Engineering Research Council of Canada grant. The funders had no role in the study design, data collection and analysis, decision to publish, or preparation of the manuscript.

**Competing interests:** The authors have declared that no competing interests exist.

## Results

We find a significant effect of familiarity and setting but not of age or musical training on recognition performance as measured by *d'*. More specifically, performance is best for the familiar, tonal piece, moderate for the unfamiliar tonal piece and worst for the unfamiliar atonal piece. Performance was better in the live setting than the lab setting.

## Conclusions

The absence of an age effect provides encouraging evidence that music's diverse cues may encourage cognitive scaffolding, in turn improving encoding and subsequent recognition. Better performance in an ecological versus lab setting supports the expansion of ecological studies in the field.

## Introduction

The association between aging and declining memory encoding and retrieving abilities is well documented [2–5]. Interestingly, there has been much less research on how aging impacts memory for music, however, interest in this topic has been growing [6–18]. One reason for the growing interest in music memory in aging is because of the growing interest in using music as an intervention for maintaining cognitive function with age [19–32]. One little explored area of musical learning and memory is how it functions in a naturalized listening environment like a symphony concert. In this study, we explore the effects of age and familiarity of memory for new music at a live symphony concert compared to a similar in-lab to test to see if a live concert setting may affect memory performance, particularly in older adults.

### Memory in older adults

A common complaint in older adults is trouble with their memory, especially for new information (i.e., encoding; [33, 34]). Current knowledge about normal aging and changes in memory identify a divide between memory tasks that are unaffected by aging and those that are. For this study, investigating familiarity for new music in an ecological listening situation, we focus on the processes of encoding and recognition, and semantic memory, as it underlies schematic knowledge of tonality. Memory tasks un- or less affected by aging include recognition tasks, tasks relying on well-known information, semantic knowledge and automatic processes [5, 35]. Negative age effects are seen for recall tasks and tasks that require effortful, self-initiated processing in real time [36, 37]. There is also evidence that aging negatively affects encoding and to a lesser degree retrieval processes [38–41], but not memory storage [41]. This is in line with the aging theory of inhibition, which suggests that older adults are worse at inhibiting task-irrelevant information, making perception and encoding more difficult [42]. Interestingly, other work shows that meaning, predictability or familiarity aids in visual memory for older adults [43, 44].

### Memory for music in older adults

Music is an ideal stimulus with which to study memory as it is multimodal and incredibly rich. It is also quite complex and musical memory can be quite poor [45, 46]. It appears that this difficulty cannot be accounted for by the level of processing [47] complexity of encoding as music

is not encoded using elaborative encoding strategies [48]. Instead, familiarity and well-formedness are proposed as factors with more important effects on memory for music [48]. Familiarity is one of the two cognitive processes thought to underlie retrieval, along with recollection [49]. Specifically, familiarity refers to an overall feeling of "-known-ness" without being able to place a specific context, for instance not being able to place a tune that sounds 'familiar', while recollection refers to a conscious recollection of specific details, for instance hearing a tune at a particular place and time [48]. Recollection relies on episodic memory, but episodic memory cannot be engaged if a tune cannot be identified or named (i.e. is novel; [50]. For unfamiliar music, listeners need to rely on schematic knowledge of music, expressed through the tonal hierarchy [51]. The tonal hierarchy is the hierarchical organization of pitch in music reflecting the perceptual organization of the relationships between the twelve pitches of Western tonal music's chromatic scale. Implicit knowledge of the tonal hierarchy generates expectations about what can come next and what is grammatically 'correct' [52] in the music, helping narrow possibilities and ease encoding. Importantly, this knowledge does not rely on formal musical training but both implicit knowledge [51, 53] and increased exposure (i.e. age; [54, 55]) to refine and strengthen tonal representations, supporting the encoding of novel material. The second factor proposed to have an important effect on musical memory is well-formedness. This is another way of expressing that intervals that follow the rules of tonality help ease encoding.

In terms of types of memory, existing literature on memory for music in normal aging suggests that overall explicit memory, such as in a recognition task suffers, while implicit memory, such as in a preference rating task is preserved [11, 35]. However, schematic knowledge of musical structure, accumulated over time in older adults, may help ease encoding and recognition [5, 15, 56].

## Memory for music in older adults with musical training

The effect of musical training on recognition memory in older adults is unclear. One might expect musical training to minimize any age effects due to the multimodal nature of music (visual, auditory, motor) activating rich domain knowledge [16]. Indeed, some studies propose a skill-invariance hypothesis to explain apparently conflicting memory performance as a function of age and experience. Specifically, that age effects dominate when general-purpose cognitive mechanisms are needed to perform the musical task at hand, while experience effects dominate when music-specific knowledge is needed to perform the task [1, 12]. While Meinz [16] found no interaction between age and musical expertise on memory performance or perceptual speed, "there was some evidence that greater experience on the part of older adults attenuated the age-related effects on performance of relevant cognitive tasks, especially the memory tasks." [16].

A review of the literature on memory for melodies [48] suggests that experience effects are found in short-term recognition tasks [57, 58] or where fine-grained pitch processing is required such as detecting non-contour violating variations on a theme [12]. Others find no effect of training [12, 45, 50, 59, 60]. Another, more recent, review [61] suggests that musical training is beneficial, where musicians hold more items in memory and for longer than non-musicians [62]. The link between musical training and working memory in general seems to be stronger than the link between musical training and memory for music specifically [63–65]. This advantage may start at the perceptual level, as auditory streams persist in echoic memory longer in musicians [66].

In summary, the potential benefit of musical training on memory for music in older adults seems dependent on both the music itself (i.e. complexity, tonality) and the type of task (i.e.

short- or long-term, recognition or recall). Meanwhile, older adults generally perform worse than younger adults on remembering complex, unfamiliar music in short-term and recognition tasks but not simple, familiar music in long-term memory tasks [1]. Additionally, according to intervention-based training studies, the benefit may be mediated by improving working memory in general [22, 26–29, 32, 67, 68].

## Speech-in-noise in older adults

Another common complaint in older adults is difficulty understanding speech in noisy environments [69], which can lead to social isolation [70], and potentially cognitive decline [71]. Along with auditory stream segregation and attention, memory–especially working memory– is an important component to understanding speech in noise [69]. While speech perception suffers in old age, music perception does not seem to be affected, as evidenced by the large percentage of older adults who still enjoy music on a daily basis [72, 73]. Therefore, the theoretical framework for this study is cognitive scaffolding [74], specifically the potential for cognitive skills maintained in music listening being used as a cognitive scaffold upon which to build speech in noise listening skills [75]. Having previously found that sequential auditory streaming by loudness and timbre is maintained in older adults [76], in addition to auditory streaming by pitch [77], this study focuses on memory, specifically the encoding of new information and its subsequent recognition [18].

## Current study

Music can be considered an engaging, active task, particularly when listeners are given a task to complete in relation to the music itself, such as identifying a theme each time it occurs in a piece of music. In this study, attendees of the Newfoundland Symphony Orchestra's 2022–2023 Season Masterworks 3 concert–and a separate sample of participants from the general population for the lab-based study identified when they heard the theme for three pieces of music manipulated for familiarity. Different from a feeling of "known-ness", in this study familiarity was operationalized along two dimensions: tonality (schematic memory) and novelty (veridical memory). The first and most familiar piece was Eine Kleine Nachtmusik by W. A. Mozart, both tonal and previously heard to most. It also served as a control to ensure participants understood the task. The second and third pieces were commissioned from students of Memorial University of Newfoundland's School of Music. One was tonal and the other atonal so that the second piece, Pirate Waltz, was tonal and novel and the third piece, Unexpectedly Absent, atonal and novel. In this way, the study includes three levels of familiarity considering both schematic (tonal vs atonal) and veridical (previously heard vs novel) forms. Participants completed a demographic questionnaire, including information on musical background, a hearing self-assessment [78] and two short cognitive tests.

As an effortful task requiring active processing (of the incoming sound), the suppression of irrelevant information (ambient noise; non-melodic content to a certain extent) and the encoding of new information (the theme), it would be expected that older adults would perform worse than younger adults at such a task. However, enough time to 'digest' the information presented (i.e. later repetitions of the theme [6, 10]) and degree of familiarity may mitigate these challenges. Combined with work demonstrating that older adults rely on tonal structure more than note-to-note interval information to process music [55], we expect older adults to perform a theme-identifying musical memory task better for familiar, tonal music than for unfamiliar, atonal music [7, 8, 15, 79]. It is possible that access to tonal information offsets the typical encoding difficulty observed in older adults, where older adults would perform at a similar level, or better, than younger adults for familiar, tonal music and similarly

but not better for unfamiliar tonal music [8, 15]. In other words, we predict an interaction between age and familiarity for melody recognition. As this study includes age as a key variable, we also measure confounds related to cognitive fitness including social and physical activity and education [80]. We expect these, and musical background to correlate positively with task performance.

## Materials and methods–live concert

### Participants

Participants were attendees of the Newfoundland Symphony Orchestra's Masterworks 3 concert from their 2021–2022 season which took place on May 20th, 2022. Those interested in taking part in the study provided written informed consent in accordance with the Interdisciplinary Committee on Ethics in Human Research at Memorial University of Newfoundland and Labrador (20210452-ME). Participants could enter into a draw for one of ten $25 cash gift cards.

Concert attendees might have included locals, settler or Indigenous, immigrants and tourists, though we did not collect this specific information as part of our demographic questions. We did however collect information about musical background and age. For additional context, Newfoundland and Labrador has a rich musical culture comprising many genres, including widespread and popular Irish/Newfoundland folk; it is more difficult to find someone without any musical performance experience, formal or informal, than with. This province also has the largest aging population in Canada, with those born between 1945–59 (i.e., 'baby boomers') making up 31.1% of the total population [81].

Due to the live setting of this study, the dataset is incomplete. We received experimental data from 108 participants. Of these, 92 completed the questionnaires, which included the key demographic detail of age. The mean age of this group was 46.8 (SD = 17.4, range = 18–86). Of these, 31 also completed the cognitive tests. The mean age of this group was 45.7 (SD = 14.1, range = 19–64). Table 1 shows the demographic distribution of our participants for each data subset by decade of age, including mean years of education, hours of activity per week, hours of socializing per week, hours listening to music per week, years of formal training, adapted hearing screening inventory and mean letter-number sequencing, Trail Making A and Trail Making B scores.

**Table 1. Demographic distribution of participants for data subsets in the live concert setting, before removal any participants.** Questionnaire = clicker and questionnaire. Cognitive Tests = clicker, questionnaire and cognitive tests. Response ranges for Hours of activity, socializing and listening to music were translated into a single integer before averaging, in this case the high end of the range.

| Age Range | Questionnaire | | | | | | | Cognitive Tests | | | |
|---|---|---|---|---|---|---|---|---|---|---|---|
| | # of participants | Years of education | Hours of activity | Hours of socializing | Hours listening to music | Years of formal musical training | Hearing Abilities self-assessment | # of participants | Letter-numbers sequencing | Trail Making A | Trail Making B |
| <20 | 5 | 15.0 | 4.0 | 5.0 | 7.0 | 7.7 | 22.8 | 1 | 12.0 | 16.6 | 30.1 |
| 21–30 | 19 | 17.8 | 7.3 | 6.6 | 5.3 | 7.4 | 21.2 | 6 | 9.3 | 18.9 | 40.1 |
| 31–40 | 14 | 18.9 | 6.0 | 5.1 | 7.4 | 9.4 | 21.8 | 7 | 12.0 | 17.1 | 33.7 |
| 41–50 | 13 | 17.4 | 5.0 | 4.7 | 4.5 | 7.9 | 22.4 | 4 | 13.0 | 19.3 | 37.0 |
| 51–60 | 20 | 18.3 | 6.3 | 6.6 | 7.9 | 6.6 | 24.3 | 10 | 10.1 | 25.7 | 49.6 |
| 61–70 | 12 | 18.2 | 6.3 | 5.0 | 4.7 | 4.6 | 25.6 | 5 | 10.4 | 27.6 | 46.5 |
| 71–80 | 9 | 18.4 | 7.0 | 6.0 | 5.3 | 6.0 | 22.0 | 0 | - | - | - |
| 80+ | 0 | - | - | - | - | - | - | - | - | - | - |

## Materials

**Pieces of music.** Three pieces of music were used in this study. Eine Kleine Nachtmusik by W.A. Mozart was a control piece likely familiar to participants to ensure the task was understood. The theme was identified as the first 8 measures of the piece. The piece was performed with no repeats, resulting in a total of two iterations of the theme. Two 5-minute experimental pieces were commissioned from students of Memorial University of Newfoundland and Labrador's School of Music. Their instructions were to include 10–12 exact iterations of a chosen theme and that some should be easy to perceive, some difficult, and some of medium perceptibility. This gradient was primarily achieved through texture and was introduced to avoid floor or ceiling effects. For example, an "easy" theme could be performed solo, or by a prominent instrument with little accompaniment. A "hard" theme is often played by an instrument in its weak range among the entire orchestra playing loudly. Composers were free to include secondary themes and foils, or part of the theme. One piece was composed in the tonal system and another in the atonal system. The final version of each piece included 11 themes. Figs 1 and 2 illustrate each piece's theme, as well as an example of the theme in an easy, medium and difficult context. Audio examples of each context can be found on the project's OSF page (OSF | Age and familiarity effects on musical memory).

**Clickers.** 150 LCD clickers by Turning Technologies Canada (Turning Technologies - Wikipedia; no longer exist as a company) were rented from Data on the Spot (Home – Data On The Spot) along with a receiver. Clickers were attached to lanyards and were numbered 1–150 with stickers on the back.

**Questionnaire.** The questionnaire contained two parts. The first part collected demographic information including age, gender, education, hours of physical activity, hours of socializing, hours actively listening to music, familiarity with Eine Kleine Nachtmusik by W.A.

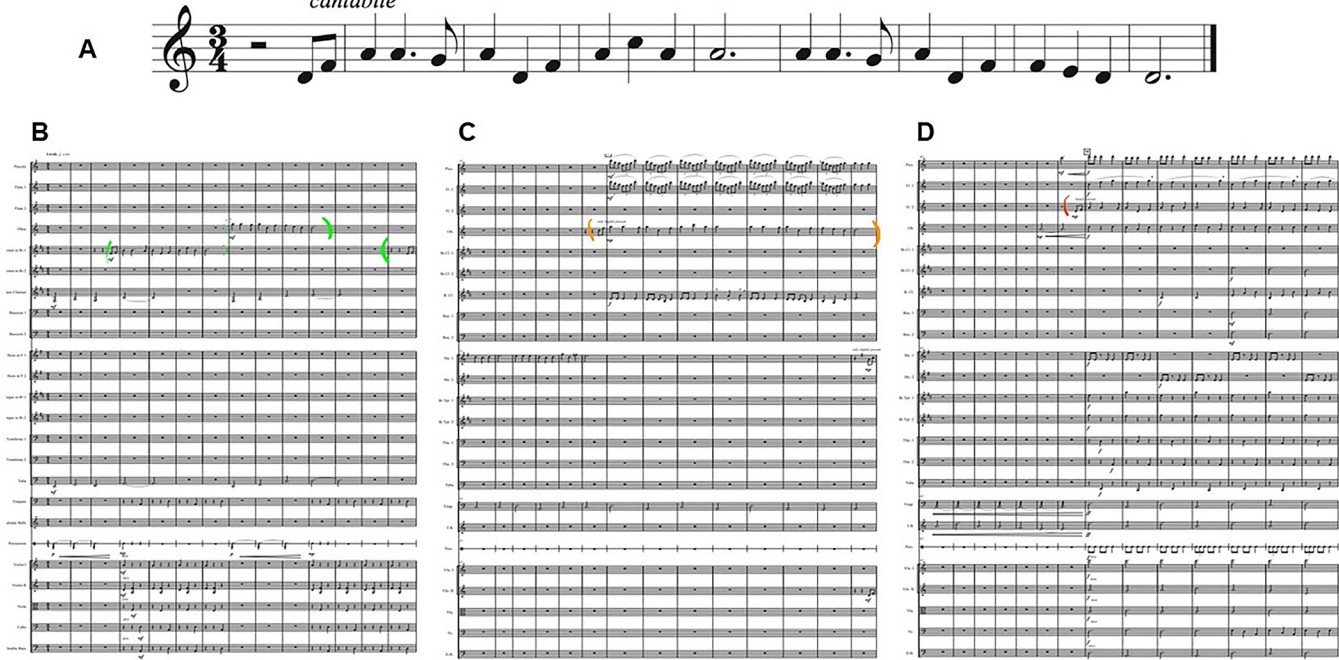

**Fig 1.** Theme (A) and sample iterations of the theme in an easy (B), medium (C) and hard (D) context for Pirate Waltz, the tonal, unfamiliar piece.

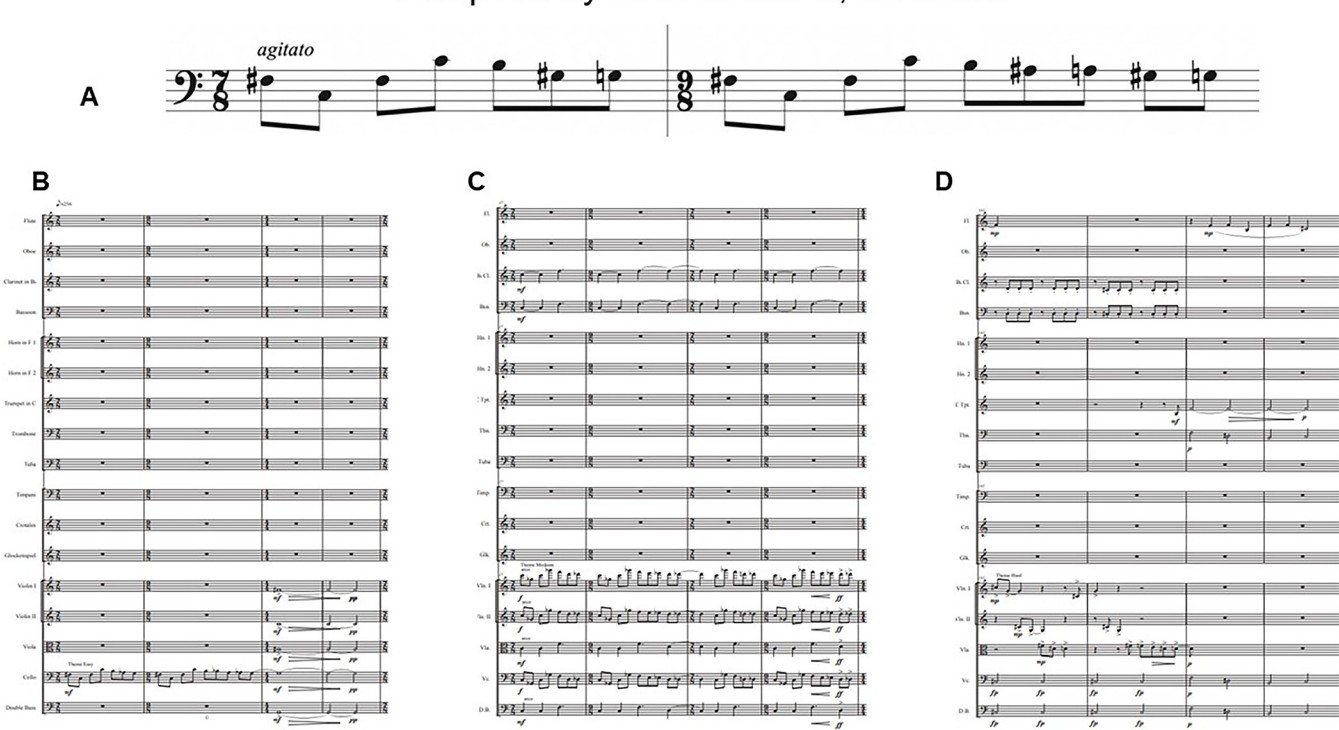

**Fig 2.** Theme (A) and sample iterations of the theme in an easy (B), medium (C) and hard (D) context for Unexpectedly Absent, the atonal, unfamiliar piece.

Mozart, and details about musical training including years of non-formal musical training, years of formal musical training, whether or not participants were currently practicing, and their primary instrument. The second part is adapted with permission from The Hearing Screening Inventory [78] and consisted of 12 items describing common situations related to hearing. For example, "Are you ever bothered by feelings that your hearing is poor?" or "Can you hear the water boiling in a pot when you are in the kitchen?", for which answer options were "never", "seldom", "occasionally", "frequently" or "always". Other questions were statements, such as "Overall, I would judge my hearing in my right ear to be. . ." or "Overall, I would judge my ability to make out speech or conversation to be. . .", for which answer options were "good", "average", "slightly below average", "poor" or "very poor".

**Cognitive tests.** Two cognitive tests were selected for their relevance to working memory and general cognitive functioning, and brevity due to the live concert setting. These are the letter-number sequencing subtest from the Weschler Adult Intelligence Scale, Third Edition [82] and the Trail Making Test, A and B, a common neuropsychological instrument [83]. In the letter-number sequencing test, participants are read a series of letters and numbers, a "trial". Participants are asked to recall the numbers first in ascending order and then the letters in alphabetical order. For example, "3 –W– 5" would be "3–5 –W". Trials begin with two letters or numbers and gradually increase to eight. Each level has three trials; the test stops when a participant scores 0 on all three trials in a level. The total possible score is 21. In the Trail Making Tests, participants connect circles containing numbers 1–25 (A) in ascending order or circles containing letters and numbers in alternating ascending and alphabetical order (B; e.g. 1-A-2-B-3-C). Participants are timed in seconds; performance will be interpreted in the context of normative data by age and education [84]. All materials can be found on the project's OSF page (OSF | Age and familiarity effects on musical memory).

## Procedure

Given that this study took place in a live concert setting, the procedure is non-standard. The research team was set up in the lobby of the concert hall at two tables with one team member each, and the rest floating around. Research team members were members of the Cognitive Aging and Auditory Neuroscience laboratory, as well as volunteer recruited from Memorial University of Newfoundland and Labrador's Human Neuroscience Research Group and School of Music. All volunteers were trained for at least one hour to familiarize themselves with the protocol of the study and conducting the cognitive tests. At the concert, anyone interested in taking part in the study could approach a member of the research team to collect a study packet (two copies of the informed consent form, the questionnaire, written instructions and a prize draw ticket) and a clicker, provided the informed consent form was read and signed. The questionnaires were pre-numbered so that participants received matching questionnaires and clickers. Questionnaire and cognitive tests could be completed at any time before the concert, during intermission, or after the concert. Questionnaires could be completed by participants themselves. To complete the cognitive tests, participants needed to find a member of the research team, who would take them to a relatively isolated spot around the building to complete the tests. This was not ideal, as the surrounding area was noisy; however, given the live concert environment, this was the context we had to work with.

The study pieces were programmed at the beginning of the concert. Before the music started, the first author introduced the research lab, the study, and gave participants oral instructions from the stage as follows:

"Before each study piece, you'll hear a theme 3 times. The first time, the whole orchestra will play it. The second and third times, it'll be a soloist. Your job is to click a button–any button–on the clicker when you hear the theme. Remember to listen for the notes, not the instrument who's playing. Also–this one's important–make sure to wait until you hear the whole theme before clicking; sometimes the composer uses only the beginning but then goes somewhere else with it. Last thing: try as much as you can to keep your clicker hidden, maybe under a scarf, jacket, or in your hands. Ok, I hope that's clear. Click a button when you hear the theme. Thank you for taking part in this live concert study! Enjoy the music!"

The orchestra performed Eine Kleine Nachtmusik, Pirate Waltz, and finally Unexpectedly Absent, playing the theme three times before each piece [9], once by the entire orchestra in unison and two more times by a soloist. For Eine Kleine Nachtmusik, the soloists' instruments were a violin and a double bass. For Pirate Waltz, the soloists' instruments were a clarinet and a horn. For Unexpectedly Absent, the soloists' instruments were a cello and a trumpet.

## Data collection

Clickers were operated using Turning Technology software. This software is designed for voting and is not specifically adapted for the application required in this study. The software was implemented to accept clicks in intervals of 5 seconds (i.e. a new poll opened every 5 seconds). The software reported the computer's time (hh:mm:ss) at which a slide was viewed, the computer's time at which polling opened, the computer's time at which polling closed, and for each clicker device, the time in seconds since polling opened when a button was pressed. Indeed, no more than one click per 5 second window was recorded for any device.

## Data processing

**Clickers.** Clicker data was converted from PDF documents exported by the software to Excel spreadsheets by the research team. The Turning Technology software produced a session log, containing the polling opened and polling closed time of each slide, and a participant

response time log, which included the clicker (participant) number and the response time of each click for each slide in seconds (to the hundredth millisecond). Session log data was stored in one spreadsheet and response times in another to the nearest second (hh:mm:ss), where the common variable was the time range of a slide (e.g. 1:30–1:35). All data was encoded by one team member and verified by another. Given the start time of the piece and the start time of each poll (slide), it was possible to determine the time (mm:ss) of each click from the beginning of the piece for each device (participant).

**Questionnaire.**   Questionnaires were digitized to Excel spreadsheets. One team member encoded the data and another team member checked the data. The hearing abilities questionnaire responses were converted to numbers, where questionnaires responses were encoded as either 1 through 5 from left to right so that "never" and "good" were equivalent to 1 and "always" and "very poor" were equivalent to 5 for questions 1, 5, 6, and 9–12 and the opposite for questions 2, 3, 4, 7 and 8. Where one or two questions were unanswered, data was interpolated as the mean of all answered questions; 0.5% of the data was interpolated in this way. Responses to each question were totaled for an overall hearing abilities self-assessment score, where the higher the score, the worse the hearing ability.

**Cognitive tests.**   Cognitive test results were digitized to Excel spreadsheets. One team member encoded the data and another checked the data. A score out of 21 was recorded for the letter-number sequencing test, where the higher the score the better the performance. Time in seconds and milliseconds was recorded for the Trail Making A and B tests, where a shorter time represents better performance. Tester initials were also recorded for each participant in order to test for any experimenter effects. Trail-making times were compared to normative data stratified by age and education [84] to screen out participants with abnormally long times, which may indicate some type of cognitive impairment; one participant from the 51–60 age range was removed.

## Data analysis

**Ground truth.**   It was necessary to generate a ground truth in order to identify whether a click was a hit or a false alarm. To do so, the start and end time (mm:ss) of each iteration of a theme was recorded based on a video recording of the concert. Once video time markers were recorded, the start time of the piece was subtracted from each theme time to arrive at a start and end time from the start of the piece. To determine a "hit" window, 2 seconds were added to each theme's end time, corresponding to approximately 2–4 measures, depending on the tempo of the piece. Given participants were instructed to wait until the end of the theme to click, it was necessary to extend the window in which a click would be counted as a hit. However, it was also important to consider that participants may click early (i.e. during the theme; many did) and to also consider these correct identifications of the theme. Clicks from each device, converted to mm:ss format in data processing, can be compared to the hit windows identified in the ground truth.

**Signal detection.**   With hits and false alarms generated from the ground truth and clicker data, these were converted to $d'$ using the R package *psycho* [85]. In 0.03% cases, there were two hits recorded for a theme; these were recoded as a single hit. In order to calculate $d'$, the number of possible hits and possible false alarms is necessary; $d'$ was calculated for each participant at each difficulty level, for each piece. There were 2, 11 and 11 possible hits for Eine Kleine Nachtmusik, Pirate Waltz and Unexpectedly Absent, respectively. These can be broken down to: 2 easy themes and no medium or hard themes for Eine Kleine Nachtmusik; 4 easy, 4 medium and 3 hard themes for Pirate Waltz; and 3 easy, 3 medium and 5 hard themes for Unexpectedly Absent. Possible false alarms were the number of theme-length segments that

were not themes, calculated as the number of non-theme measures divided by the length of the theme, in measures. There were 42, 27 and 30 possible false alarms for Eine Kleine Nachtmusik, Pirate Waltz and Unexpectedly Absent, respectively. The same number of possible false alarms was used for each difficulty level, as we considered it misleading to label a differently difficult theme a non-theme.

Participants who did not correctly identify any themes in Eine Kleine Nachtmusik were considered to not have understood the task and were excluded from further analysis. Twenty-two participants were excluded in this way, 6 in the 40–51 age range and 1–3 in each of the remaining age ranges.

## Materials and methods–lab setting

### Participants

Participants were members of the university and general community, who may have included locals, settler or Indigenous, and immigrants, though this demographic information was not collected. These participants were specifically *not* attendees of the live concert event and took part in the study between August 25th and September 22nd, 2022. Due to the controlled lab setting, complete datasets were collected from 31 participants. These participants were recruited to roughly match the number of participants in each age group from the live study, where an age group is a decade (starting from 20s), and to extend into the 70s and 80s age groups (which are not represented in the live study full dataset). The mean age of participants in the lab study was 50.7 (SD = 20.3, range = 22–84). Table 2 shows the demographic distribution of our participants for each data subset by decade of age, including mean years of education, hours of activity per week, hours of socializing per week, hours listening to music per week, years of formal training, adapted hearing screening inventory and mean letter-number sequencing, Trail Making A and Trail Making B scores. Participants provided written informed consent in accordance with the Interdisciplinary Committee on Ethics in Human Research at Memorial University of Newfoundland and Labrador and were compensated $10 for their time.

### Materials

The materials were the same as in the live concert event.

### Procedure

Given that this study took place in a lab setting, the procedure was standardized and slightly modified. Participants were invited into the lab, read the informed consent form and were

**Table 2. Demographic distribution of participants in the lab setting, before removal of any participants.** Response ranges for Hours of activity, socializing and listening to music were translated into a single integer before averaging, in this case the high end of the range.

| Age Range | Questionnaire | | | | | | | Cognitive Tests | | |
|---|---|---|---|---|---|---|---|---|---|---|
| | # of participants | Years of education | Hours of activity | Hours of socializing | Hours listening to music | Years of formal musical training | Hearing Abilities self-assessment | Letter-numbers sequencing | Trail Making A | Trail Making B |
| <20 | 0 | - | - | - | - | - | - | - | - | - |
| 21–30 | 8 | 17.8 | 4.5 | 6.3 | 7.9 | 2.5 | 34.6 | 12.6 | 19.5 | 39.9 |
| 31–40 | 4 | 20.5 | 4.0 | 4.5 | 4.0 | 3.1 | 33.5 | 13.5 | 18.3 | 34.1 |
| 41–50 | 3 | 20.0 | 5.3 | 4.3 | 3.0 | 2.0 | 38.0 | 12.3 | 23.3 | 20.8 |
| 51–60 | 4 | 18.7 | 6.0 | 4.5 | 6.8 | 6.3 | 37.8 | 12.0 | 30.1 | 55.9 |
| 61–70 | 9 | 18.4 | 7.8 | 6.4 | 5.7 | 5.1 | 37.6 | 11.4 | 24.5 | 56.6 |
| 71–80 | 1 | 20.0 | 10.0 | 9.0 | 10.0 | 12.0 | 32.0 | 4.0 | 53.3 | 122.0 |
| 80+ | 2 | 12.0 | 5.5 | 4.0 | 5.0 | 35.5 | 33.0 | 9.0 | 59.8 | 94.3 |

given the opportunity to ask question before providing consent. Instructions were then paraphrased to them, emphasizing that they would be pressing the spacebar on the keyboard instead of a button on a device. They were then comfortably seated in a double-walled sound proof booth to watch a video recording of the concert, beginning at the point where instructions were delivered and ending at the end of the third piece. The camera was placed in the audience so that the orchestra was centered in the frame and took up the width of the frame. The first few rows of the audience are visible. The recording used the Newfoundland Symphony Orchestra's standard in-house equipment. Participants listened to the video over free field speakers at a comfortable level.

## Data collection

A MaxMSP (What is Max? | Cycling '74 (cycling74.com)) patch was written to collect the computer's internal clock (hh:mm:ss) at the time of the click. The experimenter clicked a mouse on a computer outside the booth at the start of each piece. Once the concert recording ended, participants were invited back into the lab space to complete the questionnaire and cognitive tests.

## Data processing

**Key presses.**   Key press and piece start times were saved in a text file. Using the *lubridate* R package [86], key press times were tared to the start of Eine Kleine Nachtmusik.

**Questionnaire.**   Questionnaires were digitized to Excel spreadsheets in the same way as the live concert setting. There was no missing data.

**Cognitive tests.**   Cognitive test results were digitized to Excel spreadsheets in the same way as the live concert setting. Trail-making times were compared to normative data stratified by age and education [84] to screen out participants with abnormally long times, which may indicate some type of cognitive impairment; two participants were removed, one each from the 31–40 and 51–60 age ranges. One more participant performed abnormally fast on the TM-A test as there was some confusion and it was re-done; furthermore, this participant did not consider transpositions of the theme as identifiable, which would result in an underestimate of their recognition memory. Therefore, this participant from the 31–40 age range was also removed.

## Data analysis

**Ground truth.**   The ground truth is the same as in the live concert setting except that theme start and end times were relative to the beginning of Eine Kleine Nachtmusik, instead of the beginning of each piece. This was to minimize variation due to experimenter error, where the start of each piece was initially recorded by a mouse click.

**Signal detection.**   With hits and false alarms generated from the ground truth and key press data, these were converted to *d'* using the R package *psycho* [85]. There were no double hits in this setting. The number of possible hits and false alarms is the same as in the live concert setting. One participant from the 21–30 age range was excluded due to not correctly identify any themes in Eine Kleine Nachtmusik.

**Pre-registered analysis.**   The main analysis for this project was pre-registered on OSF (OSF Registries | Age and familiarity effects on musical memory). As pre-registered, a mixed effects multiple linear regression model predicting *d'* was implemented. The model only included full data sets for better comparison between live and lab settings. All live setting participants with a missing value for age, hearing self-assessment or letter-number sequencing were excluded from analysis, as missing this information equated to not having completed the

questionnaire and/or the cognitive tests. Eighteen live setting and 28 lab setting participants were thus included in the pre-registered analysis.

We have made one change to the original planned model: familiarity with Eine Kleine Nachtmusik was removed as a predictor variable through the review process. Therefore, the model reported below had the following fixed effects: age, familiarity (piece), setting (live or lab), theme identification difficulty (easy, medium, hard), musicianship (all relevant questionnaire items), and all covariates except familiarity with Eine Kleine Nachtmusik. Categorical variables familiarity, setting, theme identification difficulty, gender, highest degree obtained and the binary 'currently practicing' were treated as factors. Categorical variables hours of activity, socializing and listening to music were recoded to be continuous. The remaining variables were continuous. All continuous variables were mean-centered and an interaction between age and familiarity was also included in the model. The random effects structure included random intercepts for each participant and random slopes as a function of familiarity (piece) for each participant.

This analysis was carried out in the frequentist framework with an alpha of .05. Significance of each predictor was determined by the $t$ and $p$ values produced by the $lm$ function using the combined lme4 [87] and lmerTest [88] R packages. Model quality was assessed using multiple $R^2$ and the overall model's $F$-statistic, as produced by the same $lm$ function.

**Exploratory analysis.** Exploratory analyses were also carried out in the frequentist framework with an alpha of .05.

## Results

### Pre-registered analysis

The pre-registered model is summarized in Table 3. Familiarity, level of difficulty, non-formal musical training and the hearing self-assessment score were significant predictors of task performance as measured by $d'$. Crucially, age was not a significant predictor, nor did it significantly interact with familiarity.

More specifically, both unfamiliar pieces predicted worse performance (M = -0.07 and M = -0.41 for the tonal and atonal pieces, respectively) than the familiar Eine Kleine Nachtmusik (M = 2.40), but the unfamiliar pieces did not differ from each other, suggesting that tonality conferred no particular advantage, or atonality a particular disadvantage in learning and recognizing a new musical theme (see Fig 3). Participants performed worse for the hard themes (M = -0.26 and M = -0.45 for Pirate Waltz and Unexpectedly Absent, respectively) than for the medium (M = 0.01 and M = -0.37 for Pirate Waltz and Unexpectedly Absent, respectively) and easy themes (M = 2.40, M = 0.05 and M = -0.41 for Eine Kline Nachtmusik, Pirate Waltz and Unexpectedly Absent, respectively). Non-formal musical training positively influenced performance (coefficient 0.56). Finally, worse hearing as approximated by the hearing self-assessment predicted declining performance (coefficient -0.31).

Though there were no age effects on task performance, age predicted performance on cognitive tasks and the number of years of musical training (see Table 4). Participants in the live setting generally performed worse on the letter-number sequencing and both Trail-Making tests than participants in the lab setting, which is unsurprising given the noisy environment in the concert hall and atrium. Participants in the live setting also had significantly less years of non-formal musical training (see Table 5) than participants in the lab setting.

### Exploratory analysis

**Refined model, larger dataset.** Given many live setting participants were excluded from the pre-registered analysis due to missing data and that neither age nor cognitive test scores

**Table 3. Summary of pre-registered linear model including coefficient estimate, coefficient standard error, t-value and p-values.** EKN = Eine Kleine Nachtmusik.

| Coefficient | Estimate | Standard Error | t-value | p-value |
|---|---|---|---|---|
| **Fixed Effects** | | | | |
| Intercept | 1.77 | 0.15 | 12.05 | < .05 |
| Age | -0.12 | 0.13 | -0.92 | .37 |
| Pirate Waltz (vs EKN) | -2.04 | 0.16 | -12.94 | < .05 |
| Unexpectedly Absent (vs EKN) | -2.37 | 0.15 | -16.11 | < .05 |
| Setting | 0.15 | 0.15 | 1.00 | .33 |
| Difficulty (Medium vs Easy) | 0.04 | 0.06 | 0.66 | .51 |
| Difficulty (Hard vs Easy) | -0.13 | 0.06 | -2.15 | < .05 |
| Formal musical training (years) | -0.15 | 0.14 | -1.09 | .28 |
| Non-formal musical training (years) | 0.17 | 0.08 | 2.14 | < .05 |
| Currently practicing music (binary) | 0.14 | 0.15 | 0.92 | .37 |
| Gender (M vs F) | 0.03 | 0.12 | 0.23 | .82 |
| Gender (O vs F) | 0.05 | 0.36 | 0.15 | .88 |
| Education (years) | 0.03 | 0.06 | 0.42 | .68 |
| Listening to music (h/week) | 0.04 | 0.06 | 0.75 | .46 |
| Activity (h/week) | -0.08 | 0.07 | -1.09 | .29 |
| Socializing (h/week) | 0.06 | 0.06 | 0.92 | .37 |
| Hearing self-assessment score | -0.21 | 0.10 | -2.10 | < .05 |
| Letter-number sequencing | 0.07 | 0.08 | 0.96 | .35 |
| Trail-making A | -0.01 | 0.13 | -0.06 | .95 |
| Trail-making B | -0.0 | 0.12 | -0.51 | .62 |
| Age * Pirate Waltz | 0.19 | 0.15 | 1.27 | .21 |
| Age * Unexpectedly Absent | 0.21 | 0.14 | 0.89 | .38 |
| **Random Effects** | **Variance** | **Standard Deviation** | | |
| Intercept | 0.33 | 0.57 | | |
| Pirate Waltz | 0.57 | 0.75 | | |
| Unexpectedly Absent | 0.48 | 0.69 | | |
| Residual | 0.13 | 0.36 | | |

Note: $\chi^2$ (21) = 129.52, $p$ < .05, marginal $R^2$ = .75

Note: reported effect of age is the marginal relationship between $d'$ and age for Eine Kleine Nachtmusik only

were significant predictors of performance, performance based on familiarity alone can be analyzed with a larger sample size. Accordingly, a model predicting task performance ($d'$) with familiarity (piece) and theme identification difficulty as fixed effects was fitted. All confound predictors included in the pre-registered model were removed due to missing data. The model had $\chi^2$ (4, 636) = 238.88, $p$ < .05 and marginal $R^2$ = 0.68. Effects of familiarity and difficulty were significant in the same pattern as the pre-registered model (see Fig 4).

**Removing Eine Kleine Nachtmusik.** In case any effect of tonality alone was masked by the effect of veridical knowledge of Eine Kleine Nachtmusik, and given it was also used as a control piece, a model predicting task performance ($d'$) with familiarity (piece) and theme identification difficulty that excluded Eine Kleine Nachtmusik was fitted. In this way, the tonality aspect of familiarity can be tested separately from the veridical knowledge aspect of familiarity. Thirty-two participants were included in this model. It produced a significant effect of familiarity ($\beta$ = -0.35, SE = 0.09, t(29) = -4.00, p < .05), where performance for Pirate Waltz (M = -0.04) was better than for Unexpectedly Absent (M = -0.38) and for theme iteration difficulty, where hard themes (M = -0.34, $\beta$ = -0.15, SE = 0.07, t (128) = -2.13, p < .05)

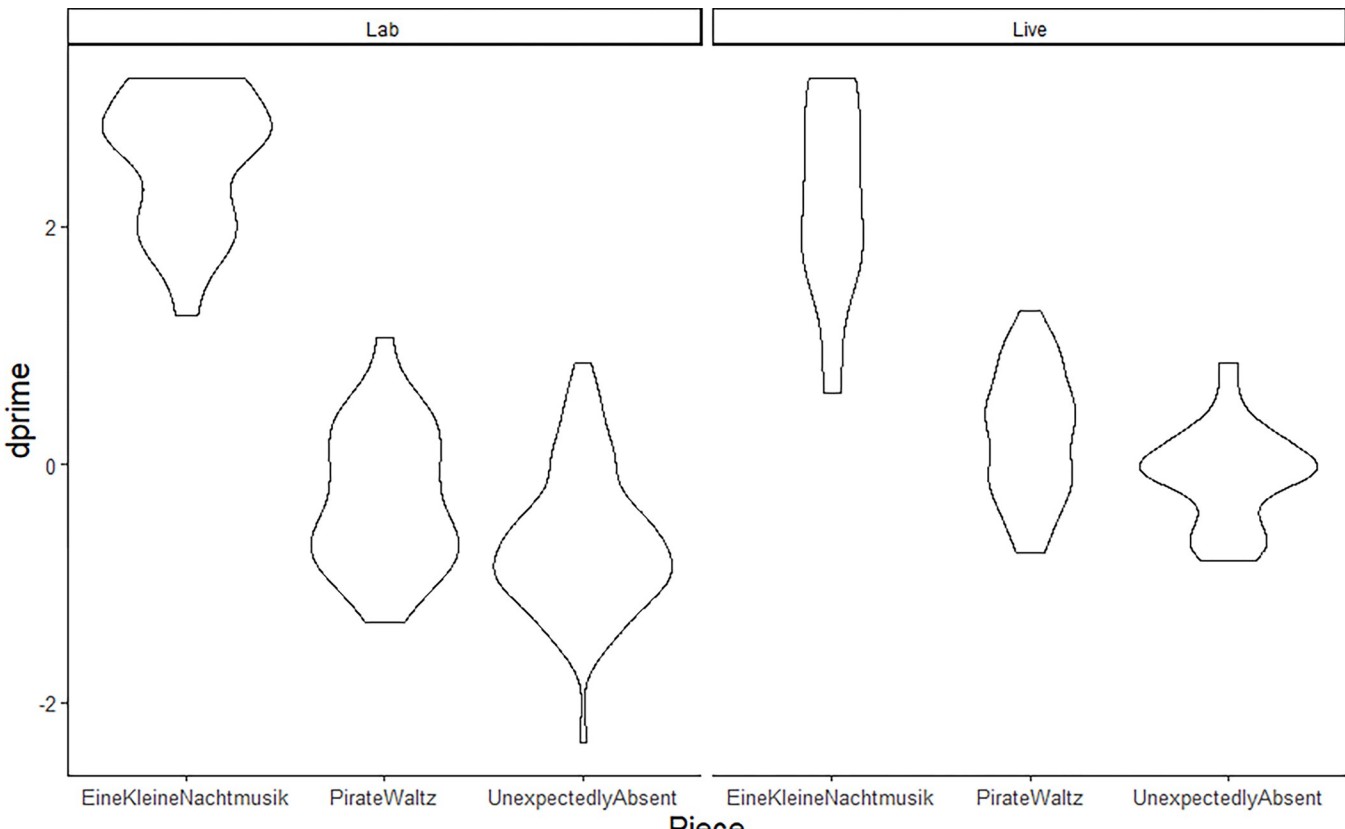

**Fig 3. Task performance as measured by d' as a function of the piece of music, where Eine Kleine Nachtmusik is familiar and tonal, Pirate Waltz is unfamiliar and tonal and Unexpectedly Absent is both unfamiliar and atonal, and setting.**

were more difficult to detect than easy themes (M = -0.19). This model had $\chi^2$ (3) = 21.51, $p <$ .05 and marginal $R^2$ = 0.10.

Given there is an effect of familiarity without the control piece, the effect of age on task performance for the unfamiliar tonal and unfamiliar atonal pieces can be re-estimated. For this model, task performance was predicted by age, familiarity and theme iteration difficulty, with an interaction between age and familiarity. Theme iteration difficulty was again a significant predictor, in the same pattern and direction as previously but critically, age was once again not a significant predictor of task performance, nor did it interact with familiarity, which was also no longer a significant main effect (see Table 6). This model had $\chi^2$ (5) = 24.82, $p <$ .05, marginal $R^2$ = .13.

**Table 4. Summary of linear models for age predicting measures of cognitive function and musical training, including coefficient estimate, coefficient standard error, t-value, p-values and model multiple $R^2$.**

| Outcome Variable | Estimate | Standard Error | t-value | p-value | multiple $R^2$ |
|---|---|---|---|---|---|
| Formal musical training (years) | 0.16 | 0.03 | 4.49 | < .05 | .07 |
| Non-formal musical training (years) | 0.21 | 0.05 | 4.52 | < .05 | .07 |
| Currently practicing music (binary) | 0.00 | 0.00 | 1.12 | .26 | .00 |
| Letter-number sequencing | -0.05 | 0.01 | -4.92 | < .05 | .08 |
| Trail-making A | 0.39 | 0.03 | 12.07 | < .05 | .36 |
| Trail-making B | 0.77 | 0.06 | 13.17 | < .05 | .40 |

**Table 5. Summary of linear models for setting (lab or live) predicting measures of cognitive function and musical training, including coefficient estimate, coefficient standard error, t-value, p-values and model multiple $R^2$.**

| Outcome Variable | Estimate | Standard Error | t-value | p-value | multiple $R^2$ |
|---|---|---|---|---|---|
| Formal musical training (years) | -0.48 | 1.44 | -0.33 | .74 | -.00 |
| Non-formal musical training (years) | -6.20 | 1.87 | -3.31 | $< .05$ | .04 |
| Currently practicing music (binary) | -0.05 | 0.06 | -0.96 | .34 | -.00 |
| Letter-number sequencing | -1.14 | 0.41 | -2.78 | $< .05$ | .02 |
| Trail-making A | -4.82 | 1.54 | -3.13 | $< .05$ | .03 |
| Trail-making B | -8.79 | 2.84 | -3.09 | $< .05$ | .03 |

**Hit rate as a function of time in piece.** Based on evidence that listeners struggle to detect matching contours but are better at discriminating intervals given a musically filled interval [48, 89] and the recency effect [90, 91], it is possible that performance also depends on the position in time of each iteration of the theme in each novel piece. Eine Kleine Nachtmusik was not analyzed here because it only contained two iterations of the theme. To test this hypothesis, the sum of hits for each theme iteration was calculated, divided by number of participants in each setting (104 participants total), and predicted by iteration number, difficulty and familiarity in a linear model. No predictor was significant. The model had $F(11, 28) = 1.59$, $p > .05$, multiple $R^2 = .38$.

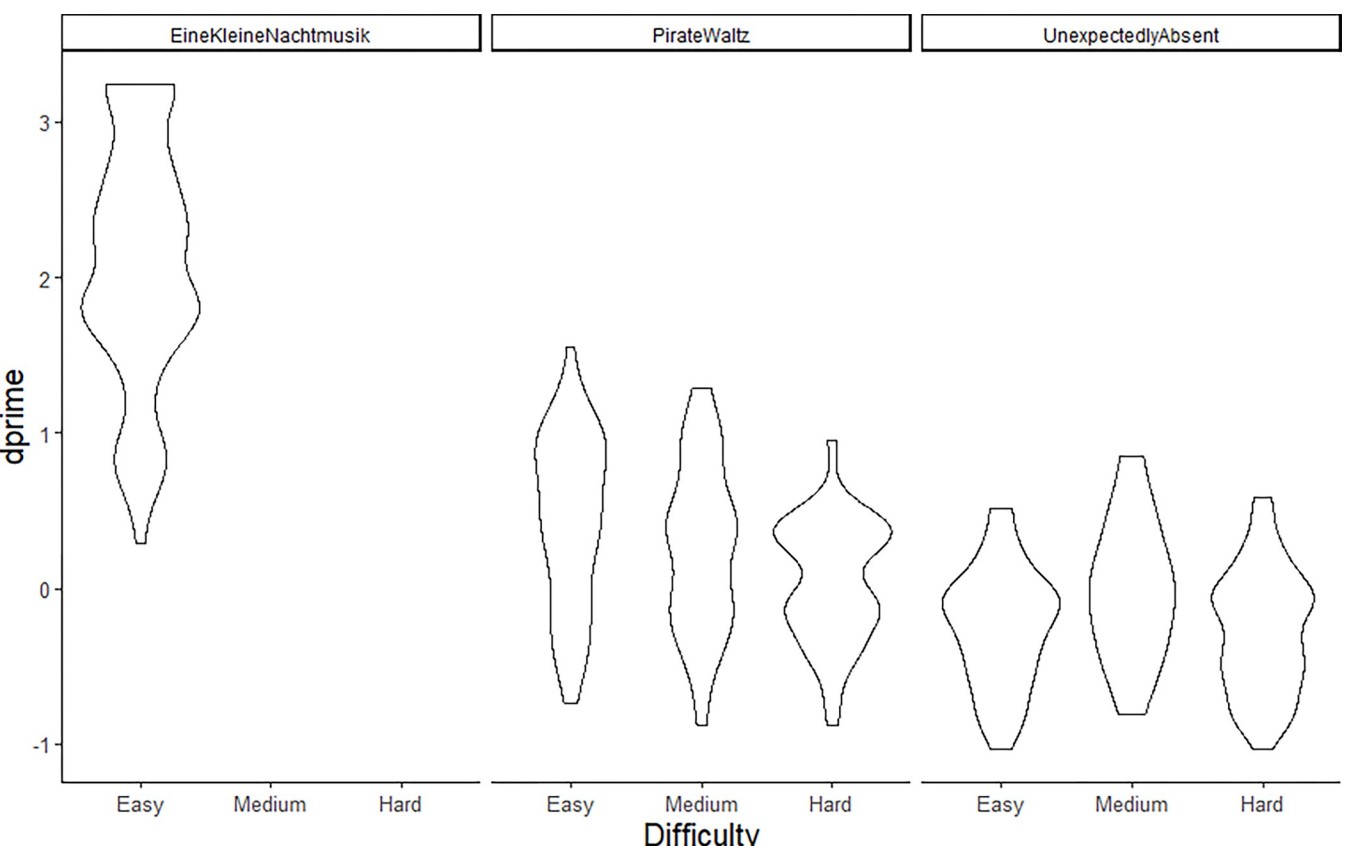

**Fig 4. Task performance as measured by d' as a function of theme iteration difficulty and familiarity for all live and lab setting participants (except those excluded in pre-processing).**

**Table 6. Summary of exploratory linear model investigating effect of age, familiarity and difficulty on performance, including coefficient estimate, coefficient standard error, t-value and p-values.**

| Coefficient | Estimate | Standard Error | t-value | p-value |
|---|---|---|---|---|
| **Fixed Effects** | | | | |
| Intercept | 0.17 | 0.24 | 0.69 | .49 |
| Age | -0.00 | 0.00 | -0.79 | .43 |
| Unexpectedly Absent (vs Pirate Waltz) | -0.16 | 0.24 | -0.67 | .51 |
| Difficulty (Medium vs Easy) | 0.05 | 0.07 | 0.65 | .51 |
| Difficulty (Hard vs Easy) | -0.15 | 0.07 | -2.13 | < .05 |
| Age * Unexpectedly Absent | -0.00 | 0.00 | -0.86 | .39 |
| **Random Effects** | **Variance** | **Standard Deviation** | | |
| Intercept | 0.20 | 0.44 | | |
| Unexpectedly Absent | 0.13 | 0.36 | | |
| Residual | 0.16 | 0.40 | | |

Note: $\chi^2$ (21) = 129.52, $p$ < .05, marginal $R^2$ = .75

Note: reported effect of age is the marginal relationship between $d'$ and age for Pirate Waltz only

**Hits and false alarms.** An analysis of hits and false alarms separately was motivated for three reasons. First, we might expect more false alarms in the context of a full musical piece compared to typical signal detection paradigms due to increased similarities across the stimulus. Incorporating melody, harmony, rhythm and timbre, there is much higher scope for mistaking any part of the music heard as sounding similar to the theme. Second, poor encoding could explain the high rate of false alarms, where participants simply do not correctly remember the theme. Third, a higher rate of false alarms has previously been reported for recognition memory in older adults while hit rates are comparable to younger adults [8, 48, 57]. This interaction is hypothesized to be due to an increased reliance on familiarity or resemblance rather than recollection or retrieval more specifically [8].Here we explore this possibility by modelling the effects of age, setting, musical training (binary 'currently practicing', hours listening to music, years of formal and non-formal training) and familiarity (piece), along with their interactions, on hit and false alarm rates calculated for each participant (92 participants total).

**Hit rate.** Familiarity (β = -0.30, SE = 0.05, t(32) = -6.56, p < .05 for Pirate Waltz and β = -0.33, SE = 0.04, t(33) = -7.52, p < .05 for Unexpectedly Absent) and age (β = -0.00, SE = 0.00, t(31) = -2.04, p < .05) but not musical training (all predictors > .05) were significant predictors of hit rate, where the model had $F$ (9, 208) = 43.13, $p$ < .05, multiple $R^2$ = .65. Specifically, mean hit rates were M = 0.33, M = 0.06 and M = 0.05 for Eine Kleine Nachtmusik, Pirate Waltz and Unexpectedly Absent, respectively. Age was negatively correlated with hit rate, with a very small model coefficient of -0.001. This is equivalent to a decreased hit rate of 0.06 for an 80-year-old participant compared to a 20-year-old participant.

**False alarm rate.** Years of formal training was a significant predictor of false alarm rate (β = -0.00, SE = 0.00, t(22) = -2.11, p < .05), but not age, familiarity or the interaction between these two, where the model had $F$ = (9, 208) = 7.45, $p$ < .05, multiple $R^2$ = .24. Years of formal training were negatively correlated with false alarm rate, with a model coefficient of -0.0007, indicating a small reduction in false alarms with increased formal musical training.

Combined, these models suggest the opposite pattern than that expected: older adults obtain slightly less hits than younger adults, but a similar number of false alarms.

## Discussion

This study explored the effects of age, familiarity on a recognition memory task for new music in a live concert and lab setting. Overall, we find no main effect of age when tasked with recognizing a theme in a piece of music, nor any significant interaction of age with familiarity, setting or musical training. Age was only a significant predictor for hit rate analysed separately. Familiarity was consistently a significant predictor, where performance was best overall for Eine Kleine Nachtmusik, a tonal and familiar piece. When removed, we also report a significant difference between the unfamiliar tonal (Pirate Waltz) and unfamiliar atonal (Unexpectedly Absent) pieces, where tonality conferred an advantage. Musical training had small positive effects, where formal musical training predicted a small reduction in false alarm rate and non-formal musical training predicted better task performance. We will now discuss these findings in the context of the literature, consider their implications for music as a cognitive scaffold for auditory rehabilitation training in older adults and the limits of generalizability of this study.

### Findings in context

**Lack of age effect.** The absence of an age effect for this recognition memory task is in line with literature on aging and memory in general [5, 35] and some music-specific literature [15]. On the other hand, this is different from studies that do find an age effect [7, 8, 11]. One of the challenges of the literature on musical memory and aging is that studies employ a range of paradigms and stimuli, where as a highly complex stimulus, small differences can have a large impact.

In this study, participants heard a theme three times then were tasked with identifying it in a piece of music in real time. A discrepancy in the age effect was reported between Bartlett et al. [7], where a melody was presented once and older adults performed worse than younger adults, and Blanchet et al. [8], where a melody was presented twice and older adults performed similarly to younger adults, both in a recognition task. Our study aligns with Blanchet et al.'s [8] results, where increased exposure–time to consolidate–seems to relieve otherwise negative effects of aging on encoding and/or recognizing new information.

The first iteration of the theme was always immediate (slight exception where there are a few bars of timpani roll and some bells in Pirate Waltz before the first theme) and additional iterations occurred separated by music-filled intervals, which Dowling [89] found aided learning. However, this was in a same-different task; in this study, all iterations were exact repetitions or transpositions of the theme, without any modifications to the interval content. We did not find any effect of iteration, either an advantage of repetition easing recognition by the end of the piece, or any primacy or recency effects.

**Degrees of familiarity.** We consistently found effects of familiarity on recognition memory. Overall, performance was best for Eine Kleine Nachtmusik, where the piece is well-known and tonal. Furthermore, the theme is clearly played by the whole ensemble. When Eine Kleine Nachtmusik was removed from analysis, we still found an effect of familiarity on performance. This effect suggests that tonality, a schematically familiar musical system, does indeed help to new musical material, at least when compared to atonality, a schematically unfamiliar system. This is expected and in line with other literature on the role of tonality or enculturation on memory [7, 8, 12, 15, 60, 79].

**Lack of setting effect.** The question of setting is particularly interesting as there is increasing interest in conducting experiments in live concert settings for increased ecological validity [92–96]. However, there are valid concerns about the loss of control that comes with a naturalistic setting that can be mitigated in-lab. Our results suggest that there is no difference, loss or

benefit, of a live setting compared to a lab setting. This is encouraging; however, further studies should continue to directly compare paradigms in live and lab settings to better evaluate and identify the benefits and challenges of live setting studies.

**The musical training non-advantage.** The weak positive results around musical training mirror a mixed literature. Overall, all significant model coefficients relating to musical training were small or very small, suggesting that even if there is a benefit or hindrance of musical training on memory performance, it likely does not have much practical impact on day to day listening. The decrease in false alarms with increased musical training suggests that musicians are more discerning of what is not a theme, though are not necessarily better at identifying themes. It is interesting that musical training did not confer an advantage for the atonal piece, where interval discrimination is more difficult without schematic knowledge of the musical system, and where previous literature suggests musicians actually have an advantage due to domain-specific knowledge [1, 12]. The richness of the stimuli, here fully orchestrated pieces of music instead of monophonic melodies as in most studies, may have mitigated learning and recognition difficulties by providing enough context cues for everyone to pick up on.

It is also possible that the lack of a musicianship effect is related to the relationship between perceptual advantages in musicians and the motor system, where it is the playing of the music that confers an advantage rather than listening alone [75, 97, 98]. In this context, it is possible that many musician participants have played Eine Kleine Nachtmusik but can never have played Pirate Waltz or Unexpectedly Absent and therefore do not have the advantage of motor processing in addition to perceptual processing.

## Music as a cognitive scaffold

As discussed in the introduction, this study was designed in the context of the theory of cognitive scaffolding [74], where cognitive skills maintained in music listening can be built upon to improve speech-in-noise perception [65, 75]. There is a general idea that active engagement in an activity like music is beneficial in old age [99, 100]. More specifically, several longitudinal training studies have investigated the effect of musical training on working memory in older adults, with positive outcomes [19, 22, 26, 27, 32]. Some work even specifically suggests changes in neural oscillations as the mechanism by which training improves working memory [61]. This study further supports the use of music in particular as a medium for cognitive maintenance and training in older adults by offering evidence that recognition memory is not affected by age in a realistic listening situation. Accordingly, music recognition could be considered a strength, onto which other aspects of memory could be scaffolded in a rehabilitation setting. For example, it may be easier to remember something if you pair a melody with it. In fact, this idea is not new. Throughout history songs have been used to transmit information orally between generations.

## Limits of generalizability

Though the lack of a detrimental effect of age on recognition memory for real music is encouraging, we must be careful of how we generalize these findings. Theoretically, we sampled from the general population, which in St. John's, Newfoundland has been increasing in diversity over the past years due to immigration and the presence of Memorial University of Newfoundland [101]. However, attending a symphony concert and having free time to participate in a lab experiment are representative of a higher socio-economic status. While the Newfoundland Symphony Orchestra makes efforts to engage with the public and be more accessible to a wider audience, it is still best to assume more wealth than less in this context. Higher socio-economic status has implications for other demographics such as race (white), sexuality

(heterosexual) and disability (able-bodied), where global systems of power benefit those most similar to the cis-hetero white man [102]. Furthermore, immigration at the current rate is a recent phenomenon in Newfoundland and Labrador, resulting in a justified assumption that most older adults in attendance at our concert and lab studies were white. The Indigenous population of Newfoundland and Labrador is likely to be under-represented, if included at all, due to historical and ongoing systems of oppression that impoverish Indigenous populations [103, 104].

Research on the intersection of memory and race seems to be focused on memory for faces, demonstrating a same-race preference [105]. We found no work on race effects for auditory memory; however, some cross-cultural research in other musical domains can formulate some possibilities. Overall, it appears that enculturation has an effect on musical dimensions such as grouping, hierarchy, tonality and entrainment [106]. Generally, while non-Western listeners 'understand' Western tonal and their own cultural conventions, Western listeners do not. Therefore, in the context of our study, we would not expect to find an effect of race on memory performance. However, we can make some other possible connections. For example, living in a white supremacist settler state like Canada exposes a racialized person to ongoing trauma [107] and Indigenous populations additionally cope with intergenerational trauma [104]. Trauma in turn negatively affects memory processes [108]. On the other hand, racialized populations, especially Indigenous peoples, have strong oral memory traditions, which might suggest improved memory for auditory content and for music more specifically. Either way, the implications of demographic traits such as race, sexuality and ability are not known because they were not studied. Our study is likely representative of a predominantly white, middle to upper class settler population and should not be generalized to other populations.

We must also address a number of limitations on the design of this study. First, there was an unequal number of themes in the three pieces. This discrepancy was difficult to avoid as the authors could not think of any existing pieces of orchestral music that contained 10–12 iterations of an exactly transposed themes in controlled levels of textural density (levels of difficulty). However, if we had composed pieces of music with only two easy to identify themes, we likely would have produced ceiling effects, which would not be informative. We believe we have partially addressed this discrepancy in our exploratory analysis by comparing only the novel pieces to evaluate the effect of familiarity as operationalized by tonality alone.

It is also possible that participants may have taken more than 2s after the end of the theme to identify it. We were limited in our time window to the shortest amount of time before a second iteration of the theme. This was unavoidable for Eine Kleine Nachtmusik but could have been addressed in the commissioned pieces. However, it is telling that both composers chose to write two themes in short succession and is a common thing to do. It is also possible that participants identified a theme while it was still unfolding, rather than only after, as instructed. We chose to include the entire theme as part of the hit window because we assumed that many participants would be tempted to respond early regardless of the instructions. This may have inflated the false alarm rate as some of these false alarms may have been for the one or two themes that were 'lures'. However, since there is only one instance of this in Pirate Waltz, we did not explore the frequency of this in detail. Future work could either avoid lures entirely, or systematically include lures and analyze their role. Relatedly, the structure of music increases the likelihood that any particular feature or excerpt "sounds like" something else in the piece, potentially inflating the rate of false alarms. The signal detection paradigm is designed for binary stimuli, where there is a much clearer correct or incorrect response. In music, a rhythm or harmony or interval may have sounded similar to the theme, even though the target was an exact match. This may explain the low $d'$ scores overall for the novel pieces of music, where the false alarm rates greatly outnumbered the hit rates. The authors are not currently aware of any

techniques to adapt signal detection to this particular context. Our exploratory analysis separating hit and false alarm rates attempts to mitigate this performance issue somewhat, revealing an age effect on hit rate alone that may have been masked by the high rates of false alarms.

Within the confines of the design and population limitations of our study, our findings suggest that recognition memory for new music is not particularly adversely affected by age when presented live as a full piece of music. As far as the authors are aware, memory for music in older adults focuses on short excerpts and in laboratory settings. Our findings also suggest that learning and subsequently recognizing a new musical excerpt is a difficult task for all ages, and that veridical knowledge is a better predictor of performance than tonality. Finally, performance on the task was not affected by whether the study took place in the laboratory or in a live setting, which is encouraging both for more ecologically valid work and laboratory work generalizability alike.

## Acknowledgments

We acknowledge the land on which we live and work as the stolen and unceded ancestral homelands of the Beothuk, the island of Newfoundland as part of the traditional territories of the Mi'kmaq and Labrador as the traditional territories of the Inuk of Nunatsiavut and NunatuKavut and the Innu of Nitassinan. We are thankful to live and work here as settlers (SAS, SC, AF and BRZ), and immigrants (PS, ÖD).

## Author Contributions

**Conceptualization:** Sarah A. Sauvé.

**Data curation:** Sarah A. Sauvé, Praveena Satkunarajah, Stephen Cooke, Özgen Demirkaplan, Alicia Follett.

**Formal analysis:** Sarah A. Sauvé.

**Funding acquisition:** Benjamin Rich Zendel.

**Investigation:** Sarah A. Sauvé, Praveena Satkunarajah, Stephen Cooke, Özgen Demirkaplan, Alicia Follett.

**Methodology:** Sarah A. Sauvé.

**Project administration:** Sarah A. Sauvé.

**Resources:** Benjamin Rich Zendel.

**Supervision:** Benjamin Rich Zendel.

**Visualization:** Sarah A. Sauvé.

**Writing – original draft:** Sarah A. Sauvé.

**Writing – review & editing:** Sarah A. Sauvé, Praveena Satkunarajah, Stephen Cooke, Özgen Demirkaplan, Alicia Follett, Benjamin Rich Zendel.

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
