## [Decision Letter · Decision Letter 0]

27 Feb 2024

PONE-D-23-34799Age and familiarity effects on musical memoryPLOS ONE

Dear Dr. Sauvé,

Thank you for submitting your manuscript to PLOS ONE. After careful consideration, we feel that it has merit but does not fully meet PLOS ONE’s publication criteria as it currently stands. Therefore, we invite you to submit a revised version of the manuscript that addresses the points raised during the review process.

We look forward to receiving your revised manuscript.

Kind regards,

Sukru Torun

Academic Editor

PLOS ONE

Journal Requirements:

   "The author(s) disclosed receipt of the following financial support for the research, authorship, and/or publication of this article: this work was supported by BRZ’s Natural Sciences and Engineering Research Council of Canada grant."

Additional Editor Comments:

First of all, I would like to state that I find your article remarkable in terms of its potential contribution to inspiring discussions and research in the context of the real-world implications of the questions and results it addresses. As a matter of fact, I think our reviewers' thoughts were in this direction. Therefore, although it was rejected by reviewer 1, with valuable criticisms that I also found instructive, I believe that the paper deserves improvement effort and agree with reviewer 2 in recommending a major revision. In this context, I strongly recommend that you take into consideration rev1's criticisms and suggestions as much as rev2 in your revision. For example; Sentences such as 'recognition tasks are unaffected by aging' or statements such as 'it is unlikely that emotion had a significant impact in this study as all three pieces could be considered high on arousal and possibly also valence' should be supported with more convincing arguments. Ek olarak, It would be appropriate to rely as much as possible on the data reported in the discussion section and to be careful not to rely on concepts that are not closely related to the measurements made. Son olarak, I liked the Limits of Generalizability section, but there seems to be a need to add similar self-critical content that addresses the limitations of the whole study.

Reviewers' comments:

Reviewer's Responses to Questions

**Comments to the Author**

1. Is the manuscript technically sound, and do the data support the conclusions?

Reviewer #1: No

Reviewer #2: Yes

2. Has the statistical analysis been performed appropriately and rigorously? 

Reviewer #1: No

Reviewer #2: No

3. Have the authors made all data underlying the findings in their manuscript fully available?

Reviewer #1: Yes

Reviewer #2: Yes

4. Is the manuscript presented in an intelligible fashion and written in standard English?

Reviewer #1: Yes

Reviewer #2: Yes

5. Review Comments to the Author

Reviewer #1: Manuscript ID: PONE-D-23-34799

Manuscript: “Age and familiarity effects on musical memory"

Authors: Sauve, S., Satkunarajah, P., Cooke, S., Demirkaplan, O., Follett, A., & Zendel, B.R.

In the reviewed manuscript, the authors examine the effects of age and familiarity on the ability to detect a theme in a piece of music. I found the research question of great interest and liked the real-world applicability, but had some significant concerns about the design and analysis that lowered my enthusiasm for the study. My biggest concern is that the d prime values are incredibly low for the unfamiliar songs, well below chance, suggesting that people were not successful in completing this task. With an outcome measure this flawed, it is not clear that the analysis is reliable. In addition, I found the introduction to be unfocused—it wasn’t clear which concepts were important to understand and which were not. There were also not clear definitions for a lot of the concepts, which made it more difficult to distinguish. I would recommend that the authors unpack the important factors in more detail and streamline the rest. I discuss specific suggestions and other concerns in more detail below:

1. I think it is misleading to say that recognition tasks are “unaffected by aging”. Although it is definitely true that recall is more affected by aging, there is a very large literature showing that recognition can also be impacted.

2. In general, the “Memory in older adults” paragraph seemed disjointed and lacking organization. It would be helpful for the authors to focus on the most important details and highlight relevant reviews for the rest.

3. It was not clear why the authors say quality of encoding should be considered separately from familiarity and well-formedness—it is very likely that both variables would contribute to quality of encoding.

4. It was not clear until late in the introduction how the authors were operationalizing the “familiarity” variable. This was particularly confusing given the compound nature of the operationalization (both actual familiarity and tonality) as well as the discussion in the introduction of familiarity with music, more generally. Further, although the definition of familiarity as the feel of “known-ness” (relative to recollection) is not incorrect, it isn’t really what you mean by familiarity in this study. This just adds to the confusion about how familiarity is defined here.

5. Is it important for the reader to understand the nuances of well-formedness and interval information? Does this contribute to tonality? If so, this is unclear. If not, I feel like this section could be removed.

6. Why do the authors include a discussion of implicit and semantic memory? It feels like they are leading into the idea of schematic knowledge, but that is not made explicit and I am also not sure how the schematic knowledge piece fits in – is this related to the music training variable later?

7. Similar to my concern about familiarity not being operationalized, there was no definition of tonality in the introduction. How are the authors establishing tonality and how does it relate to the concepts already introduced?

8. At the very end of the introduction, the authors introduce the hypothesis that social and physical activity, education, and musical background will all correlate with task performance. This came out of no where, with no focus on these variables in the introduction. There was also no discussion about how any of these may relate to or interact with age. This does not seem like a motivated hypothesis and introduces a lot of new variables to consider (see below)

9. It was very confusing for the authors to describe the possible demographic breakdown of their sample, while not actually discussing the fact that they didn’t collect this information.

10. The authors do not explain why they used three levels of difficulty in their music. This was surprising since the “high familiarity” piece of music did NOT have these three levels, meaning that there was a significant confound of difficulty in the familiarity variable. This choice should be explained.

11. Does Table 1 include the participants who were dropped for not getting any correct in the familiar/easy condition?

12. Were there any age differences in any of the other cognitive or music experience variables?

13. Similarly, were there any differences in any measures across the two groups (in lab v. live)?

14. The in-lab sample is incredibly small and, therefore, probably highly underpowered. Given that the was an in-lab study (and not a sample of convenience) why was it so small? There also are not many older adults, which makes it harder to find age effects.

15. What were the cut-offs for excluding participants with “abnormally long times” on the trail-making task?

16. If participants were told to wait until the theme was over, 2 seconds after the end of the theme might be too short of a window, particularly for older adults. Did the authors check to see if there were “false alarms in the few seconds after the window closed?

17. It seems like there are a lot of confounds in the design, with the most familiar theme having so many fewer themes, overall, and only easy themes.

18. It is really telling that 22 participants in the live condition had to be excluded for not getting any hits in the easiest condition. What did their data look like? Did these participants not respond to anything? Were their responses late? Or is the task just too hard?

19. There were a LOT of variables in the model, which can make it difficult to identify significant effects, particularly with so few participants. Did the authors look at whether any of these variables were correlated?

20. What was the final N for the pre-registered analysis, once all exclusions were done?

21. The overall accuracy for the memory task was well below chance for the unfamiliar pieces, making me wonder if participants were even doing the task. I don’t think it is possible to draw conclusions about any predictor variables when the entire sample is performing so poorly.

22. Please report the actual p-value for all tests, not just that it is below .05. Based on the t-values, it appears that many of the p-values would just barely be under .05, which is concerning given the low power in the study.

23. The t-value for Unexpectedly Absent vs. EKN is very low despite the significant p-value. Is this a typo?

24. There are a lot of exploratory analyses and not all are sufficiently defended – What questions are the authors trying to answer with each? Why use different subsets of variables across all models?

25. In addition to the age effects being unreliable due to poor overall memory, null age effects are difficult to interpret in the absence of significant effects in other domains. It would be stronger if the authors could report age-related effects in other cognitive measures.

26. The discussion often felt like it was going beyond what could be said based on the reported data. In addition to the concerns already reported, there were also sections in which the authors introduce brand new concepts that were not related to anything they measured. For instance, they introduce the potential confound of emotion, and then end by saying that there is no way to know if and how emotion would have an effect. This is something that could have been measured or could easily be included in a future study. Similarly, the extensive discussion of potential race effects seems like a stretch given that the authors did not collect demographic data and do not know what the make-up of their sample actual is.

Reviewer #2: In this manuscript, the authors investigated the effects of age, familiarity (previously exposed & tonal, novel & tonal, and novel & atonal), and setting (live vs. experimental) on performance on a music recognition task. They found that performance on this task was best for familiar music and for participants in the live music setting. Furthermore, they did not identify any effects of age or formal musical training on task performance in their pre-registered models. The authors also present several exploratory analyses which demonstrate better task performance on tonal vs. atonal novel music and a negative age-related effect on hit rates.

This manuscript is well-written and should be commended for its use of multiple methods (lab- and concert-based experimental settings), interdisciplinary collaboration with composers, and open science practices. The strength of this manuscript lies within its overall clarity and experimental design. General concerns over some statistical choices and the reporting of multiple similar exploratory models somewhat obscures the main takeaways of the manuscript. These concerns can be addressed with a major revision.

I have some concerns over the pre-registered model which tests the primary hypotheses of the manuscript. This model included fixed effects of age, familiarity, setting, theme identification difficulty, along with 10 other covariates, and random effects and slopes for each fixed effect. I commend the authors for striving for maximal random effects structure. However, because this model did not converge, the authors moved to a simple linear regression. It is likely that this model did not converge for multiple reasons that could be addressed by the authors before moving to this linear regression model, particularly given the small sample size relative to the amount of random effects. The design of the experiment might not actually allow for random slopes estimation for some fixed effects; for instance, it is not possible for the model to estimate a random slope for the effect of setting per participant since the manipulation of setting was between-participants.

As suggested by Barr et al. (2013), the authors should iteratively simplify the random effects structure to see if models with less random effects terms converge before moving to a simple linear regression. The authors should also note that It is also important that age is mean-centered in this model (and other models with interaction terms) because it is being tested in an interaction; mean-centering all continuous variables will also assist in helping the model converge (Enders & Tofighi, 2013). Given the large number of predictor variables in this model, I’m concerned that this model is overfit for the amount of participants included. The authors should consider providing justification for reducing the amount of variables in the model (perhaps running a separate model for each experimental manipulation, see next comment) or taking a step-wise or LASSO regression (McNeish, 2015) approach (for mixed effects model, the glmmLasso package in R might work). This may also assist in model convergence. Because d’ is a repeated within-participant measure, all analyses which use this as the outcome variable should prioritize models which can estimate participant error. However, should the authors decide to maintain the simple linear regression model, they should report the variance inflation factors for each predictor variable in this model, as I’m concerned about high multicollinearity of related covariates and manipulations in that model (potentially with the performance package in R).

Relatedly, I’m not entirely sure what to make of the inconsistent results across the pre-registered model and subsequent exploratory models reported, and I’m not sure if some models are necessitated or properly motivated. For instance, there is no difference in d’ between Pirate Waltz and Unexpectedly Absent in the pre-registered model but there is a difference in the next exploratory model that uses all participants and less covariates (a similar instance is the effect of theme difficulty across levels for the first three models reported). The manuscript would benefit from a more succinct and streamlined results section; namely through removing the reports of the first three exploratory analyses. If one motivation for including the first exploratory model is the larger dataset, then perhaps in line with reducing the amount of variables / hypotheses tested in the pre-registered model, the authors could test their separate hypotheses in different models. For instance, testing the effects of 1) age interacting with piece familiarity, 2) theme difficulty, and 3) cognitive task performance on d’ in separate models (all with relevant covariates). This will reduce the amount of predictor variables in the pre-registered model, hopefully provide a bit more consistency in the reported results, and allow for a greater number of participants in these models. Furthermore, such an approach seems justified by the authors’ introduction section, as the only variable they predicted interacting with age was piece familiarity.

I don’t believe that participants’ familiarity with EKN should be included in a model which estimates d’ for all three pieces. It seems a bit more like a manipulation check (i.e., was the most familiar music manipulation actually familiar?). The authors report that this is negatively related to d’ and reason that this is potentially due to participants who may have recognized that piece that may have reported that they had never heard it. However, it’s also possible that greater familiarity with EKN could represent greater affinity towards classical music and thus worse performance on the novel (especially atonal) music (and might actually be positively related when just looking at data from the EKN). For these reasons, the authors should either remove this as a covariate in models which include d’ measures from Pirate Waltz and Unexpectedly Absent or acknowledge alternate explanations for this relationship in the limitations section.

Relatedly, I commend the authors for the thoughtful discussion on the limitations of the generalizability of the present study. However, the limitations section should also speak to potential limitations of the design or alternate explanations for the identified results (e.g., maybe the live setting participants performed better because they could see/interact with other individuals completing the same task unlike those who completed the experiment in the lab).

Participants were instructed to wait until the end of the presentation of the theme to click if they had heard it, as sometimes the composers use “only the beginning but then goes somewhere else with it.” However, the authors include identifications of the theme before it ends as a hit if it is done when that iteration of the theme does turn out to be a full iteration. Thus, it seems as though the only thing distinguishing between a hit and a false alarm in the scenario when participants click early is whether the theme unfolds to be a full iteration (and not necessarily reflective of whether or not the participants did or did not recognize the theme itself). Would it be possible to evaluate instances of early “false alarms” that are just identifications of motifs in the theme that “go somewhere else” as hits in an exploratory analysis? If this is not feasible, the authors should expand on this tension in the limitations section (and make it clearer that this would be the case of a false alarm in the methods section).

At the end of the Introduction, the authors write: “We also expect factors such as social and physical activity, education and musical background to correlate positively with task performance.” However, most of these (i.e., social and physical activity) were not motivated in the Introduction. Please add motivation for including these in the models as predictor variables or justification for their inclusion as covariates in the data analysis/results section.

For Table 1, if you are reporting the effect of age using the summary() function in R (with age interacting with familiarity), then that effect is actually the marginal relationship between d’ and age for EKN (the reference level in the model). Please clarify that in the table if that is the case (and if you would like to extract the main effect, consider using the anova() function on your model or contrast coding the factor levels of familiarity).

Please include Beta estimates, t-values, and associated p-values for all effects reported, as they are currently only reported for the pre-registered model.

Can you clarify how many participants are included in each one of these models in the results section, if they are changing across models?

The authors report that p-values have been corrected for multiple comparisons across models where appropriate, but it is not clear where specifically these corrections have been applied.

It is not clear to me how non-formal musical training was measured, and it doesn’t seem to be reported in Table 1. Please clarify.

The authors included age as a predictor variable for the model predicting false alarm rate. However, this effect is not reported.

There seems to be a typo in this sentence:

“The second part is adapted with permission from of The Hearing Screening Inventory (Coren & Hakstian, 1992) and consisted of 12 items describing common situations related to hearing.”

I found this sentence a bit confusing to read, perhaps it would be clearer to list how many easy/medium/hard themes are in each piece:

“These can be broken down to 2, 4 and 3 easy themes, 0, 4 and 3 medium themes and 0, 3 and 5 hard themes for Eine Kleine Nachtmusik, Pirate Waltz and Unexpectedly Absent, respectively.”

6. PLOS authors have the option to publish the peer review history of their article (what does this mean?). If published, this will include your full peer review and any attached files.

Reviewer #1: No

Reviewer #2: No

---

## [Author Response · Author response to Decision Letter 0]

4 Jun 2024

Response to reviewers

We would like to thank both reviewers for their detailed comments, particularly for improving the clarity of the manuscript and strengthening our analyses and the discussion of limitations of our study.

In the reviewed manuscript, the authors examine the effects of age and familiarity on the ability to detect a theme in a piece of music. I found the research question of great interest and liked the real-world applicability, but had some significant concerns about the design and analysis that lowered my enthusiasm for the study. My biggest concern is that the d prime values are incredibly low for the unfamiliar songs, well below chance, suggesting that people were not successful in completing this task. With an outcome measure this flawed, it is not clear that the analysis is reliable. In addition, I found the introduction to be unfocused—it wasn’t clear which concepts were important to understand and which were not. There were also not clear definitions for a lot of the concepts, which made it more difficult to distinguish. I would recommend that the authors unpack the important factors in more detail and streamline the rest. I discuss specific suggestions and other concerns in more detail below:

1. I think it is misleading to say that recognition tasks are “unaffected by aging”. Although it is definitely true that recall is more affected by aging, there is a very large literature showing that recognition can also be impacted.

In this sentence we are referring to the results of our study providing evidence of a lack of effect of age on recognition memory. It is not a general statement of recognition memory being unaffected by aging. We have modified the sentence to clarify this as follows on p.29:

“This study further supports the use of music in particular as a medium for cognitive maintenance and training in older adults by offering evidence that recognition memory is not affected by age in a realistic listening situation.”

2. In general, the “Memory in older adults” paragraph seemed disjointed and lacking organization. It would be helpful for the authors to focus on the most important details and highlight relevant reviews for the rest.

This paragraph, on pages 3-4, has been re-written for clarity.

3. It was not clear why the authors say quality of encoding should be considered separately from familiarity and well-formedness—it is very likely that both variables would contribute to quality of encoding.

We were referring to the complexity of encoding in the context of the levels of processing framework of memory, which was unclear. The sentence, on p. 4, now reads:

“It appears that this difficulty cannot be accounted for by the level of processing (Craik & Lockhart, 1972) complexity of encoding as music is not encoded using elaborative encoding strategies (Halpern & Bartlett, 2010).”

4. It was not clear until late in the introduction how the authors were operationalizing the “familiarity” variable. This was particularly confusing given the compound nature of the operationalization (both actual familiarity and tonality) as well as the discussion in the introduction of familiarity with music, more generally. Further, although the definition of familiarity as the feel of “known-ness” (relative to recollection) is not incorrect, it isn’t really what you mean by familiarity in this study. This just adds to the confusion about how familiarity is defined here.

We have added an operationalized definition of familiarity specific to our study, p.7:

“Different from a feeling of “known-ness”, in this study familiarity was operationalized along two dimensions: tonality (schematic memory) and novelty (veridical memory).”

5. Is it important for the reader to understand the nuances of well-formedness and interval information? Does this contribute to tonality? If so, this is unclear. If not, I feel like this section could be removed.

Well-formedness describes intervals that follow the rules of tonality; this connection is made explicit and the discussion of well-formedness and intervals is significantly reduced. The section, on p.5, now reads:

“The second factor proposed to have an important effect on musical memory is well-formedness. This is another way of expressing that intervals that follow the rules of tonality help ease encoding.”

6. Why do the authors include a discussion of implicit and semantic memory? It feels like they are leading into the idea of schematic knowledge, but that is not made explicit and I am also not sure how the schematic knowledge piece fits in – is this related to the music training variable later?

Implicit and semantic memory underpin schematic knowledge, which is very important in this context as longer passive exposure to music (as well as explicit knowledge through musical training) strengthen the tonal hierarchy, providing a benefit to older adults learning new musical content in a schema they are familiar with. We have added a sentence, p.5, explaining this connection and importance:

“Importantly, this knowledge does not rely on formal musical training but both implicit knowledge (Besson & Faïta, 1995; Krumhansl & Shepard, 1979) and increased exposure (i.e. age; Cuddy & Badertscher, 1987; Halpern et al., 2017) to refine and strengthen tonal representations, supporting the encoding of novel material.”

7. Similar to my concern about familiarity not being operationalized, there was no definition of tonality in the introduction. How are the authors establishing tonality and how does it relate to the concepts already introduced?

The tonal hierarchy is defined as following on p.4:

“The tonal hierarchy is the hierarchical organization of pitch in music reflecting the perceptual organization of the relationships between the twelve pitches of Western tonal music’s chromatic scale.”

8. At the very end of the introduction, the authors introduce the hypothesis that social and physical activity, education, and musical background will all correlate with task performance. This came out of no where, with no focus on these variables in the introduction. There was also no discussion about how any of these may relate to or interact with age. This does not seem like a motivated hypothesis and introduces a lot of new variables to consider (see below)

We appreciate that these variables were not discussed before. We do not think it is necessary to go into much detail, as they are often considered as confounds in aging research. The last sentences of the introduction, on p. 8, now read:

“As this study includes age as a key variable, we also measure confounds related to cognitive fitness including social and physical activity and education (Tucker & Stern, 2011). We expect these, and musical background to correlate positively with task performance.”

9. It was very confusing for the authors to describe the possible demographic breakdown of their sample, while not actually discussing the fact that they didn’t collect this information.

We clarify that we did not collect some of the demographic information mentioned, on p.9.

“Concert attendees might have included locals, settler or Indigenous, immigrants and tourists, though we did not collect this specific information as part of our demographic questions. We did however collect information about musical background and age. For additional context, Newfoundland and Labrador has a rich musical culture […]”

However, as musically engaged residents of the province and St. John’s in particular, our knowledge of the local population and symphony attendance is valid and relevant, which allows us to make informed assumptions about the makeup of the concert attendees.

10. The authors do not explain why they used three levels of difficulty in their music. This was surprising since the “high familiarity” piece of music did NOT have these three levels, meaning that there was a significant confound of difficulty in the familiarity variable. This choice should be explained.

We explain this choice as follows on p. 10:

“This gradient was primarily achieved through texture and was introduced to avoid floor or ceiling effects.”

It is difficult, if not impossible, to find a highly known existing piece of music with this level of formal manipulation included in it. Similarly, if the two composed pieces of music only had two easily identifiable themes, we might expect a ceiling effect. This confound is now also better addressed in the discussion.

11. Does Table 1 include the participants who were dropped for not getting any correct in the familiar/easy condition?

Yes, it does. This is now clarified in the table captions.

12. Were there any age differences in any of the other cognitive or music experience variables?

Yes, there were significant effects of age on all three cognitive tests. Performance on LN decreased and performance on the TM-A and TM-B increased with age. Years of formal and non-formal musical training increased with age, which is unsurprising, but age did not predict whether a participant was currently practicing. These results are now summarized at the end of the pre-registered analysis section and in Table 4, on pages 21-22.

13. Similarly, were there any differences in any measures across the two groups (in lab v. live)?

Yes, live participants performed worse on cognitive tests and had less years of formal musical training. These results are now summarized at the end of the pre-registered analysis section and in Table 5, pages 21-22.

14. The in-lab sample is incredibly small and, therefore, probably highly underpowered. Given that the was an in-lab study (and not a sample of convenience) why was it so small? There also are not many older adults, which makes it harder to find age effects.

We explain the sample size as follows on p. 17:

“These participants were recruited to roughly match the number of participants in each age group from the live study, where an age group is a decade (starting from 20s), and to extend into the 70s and 80s age groups (which are not represented in the live study full dataset).”

In our models, age is a continuous variable, so if there is an effect of age, we should be able to identify it given a roughly similar distribution of participant ages in each decade between 20 and 80.

15. What were the cut-offs for excluding participants with “abnormally long times” on the trail-making task?

The cut-offs are taken from normative data stratified by age and education cited in the manuscript. We did not think it was necessary to reproduce this fairly large table, though we can include it if the reviewer finds it necessary.

16. If participants were told to wait until the theme was over, 2 seconds after the end of the theme might be too short of a window, particularly for older adults. Did the authors check to see if there were “false alarms in the few seconds after the window closed?

One of the reasons we selected this time frame was that it was the shortest amount of time between two iterations of the theme. In some cases, more than 2s after the theme was the beginning of another iteration, which would also be counted as a hit for the new iteration of that theme. Therefore, we did not check for any false alarms after the window closed.

17. It seems like there are a lot of confounds in the design, with the most familiar theme having so many fewer themes, overall, and only easy themes.

As we mention in our response to point 10, it was not possible to find an existing piece of music with 10-12 iterations of a same theme manipulated for difficulty, and only 2 easy themes in new music would have likely been too easy. These caveats are addressed in the discussion, p.31.

18. It is really telling that 22 participants in the live condition had to be excluded for not getting any hits in the easiest condition. What did their data look like? Did these participants not respond to anything? Were their responses late? Or is the task just too hard?

The participants in the live condition that did not respond to the Mozart piece did respond to the other two and appeared to perform in line with others in general. It is possible that they simply forgot to respond until later in the study; however, since we cannot know this for certain, we felt it better to stick to our exclusion criteria and remove them.

19. There were a LOT of variables in the model, which can make it difficult to identify significant effects, particularly with so few participants. Did the authors look at whether any of these variables were correlated?

Yes, we considered the correlation between variables. Aside from the two versions of the trail-making test, which had r = .84, no two variables correlated above r = .66, which we do not consider to be catastrophic to the model. Our approach is generally one of maximal fitting, which involves taking into account all possible confounds and accounting for their variance. If these confounds were removed and more variance was ascribed to the main effects we are interested in, this might be misleading and over-inflate the effect of our variables of interest.

However, based on the other reviewer’s suggestions, we have re-analyzed our data including a random effects structure, which has made some significant changes to our results and discussion sections.

20. What was the final N for the pre-registered analysis, once all exclusions were done?

Final count for live and lab participants in the pre-registered analysis are now included in the pre-registered analysis portion of the materials section on p.19, shortly before the Results begin.

21. The overall accuracy for the memory task was well below chance for the unfamiliar pieces, making me wonder if participants were even doing the task. I don’t think it is possible to draw conclusions about any predictor variables when the entire sample is performing so poorly.

This is an important point, thank you for raising it. We believe there are two possible explanations for low performance. First, music is highly thematic, unlike most signal detection tasks, which could have inflated false alarms. Second, participants may have encoded the themes poorly, being therefore unable to correctly recognize them in the context of a full piece of music, which is, as mentioned, highly thematic in general. These possibilities justify our separate analysis of hit and false alarms in the exploratory analysis, and are addressed in the discussion on p. 31-32.

22. Please report the actual p-value for all tests, not just that it is below .05. Based on the t-values, it appears that many of the p-values would just barely be under .05, which is concerning given the low power in the study.

We are aware that there are different conventions relating to reporting p-values. In this work, we follow Dienes’ recommendation from Understanding Psychology as a Science that determining significance with p-values is an absolute cut-off, and to report only greater or less than the alpha to limit the temptation to interpret anything as “almost significant”, which is not possible under frequentist statistics. Therefore, we still report p-values as greater than or less than our alpha.

23. The t-value for Unexpectedly Absent vs. EKN is very low despite the significant p-value. Is this a typo?

Yes, it was, but with a new model the value is now different.

24. There are a lot of exploratory analyses and not all are sufficiently defended – What questions are the authors trying to answer with each? Why use different subsets of variables across all models?

Upon reflection, the first exploratory analysis is simply a confirmation of results of the pre-registered model but with a larger sample size, so it has been significantly shortened. However, we do still feel that the remaining exploratory analyses are relevant. In each case, we open the section with stronger theoretical justification, p.23 and p.25. Predictors were selected for relevance to the question at hand, as per our judgment.

25. In addition to the age effects being unreliable due to poor overall memory, null age effects are difficult to interpret in the absence of significant effects in other domains. It would be stronger if the authors could report age-related effects in other cognitive measures.

As added based on comments 12 a

---

## [Editor Report · Decision Letter 1]

10 Jun 2024

Age and familiarity effects on musical memory

PONE-D-23-34799R1

Dear Dr. Sauvé,

We’re pleased to inform you that your manuscript has been judged scientifically suitable for publication and will be formally accepted for publication once it meets all outstanding technical requirements.

Kind regards,

Sukru Torun

Academic Editor

PLOS ONE

Additional Editor Comments (optional):

Dear Author(s),

I am pleased to see that your article has gained a stronger and more effective structure with the revisions you have made, taking into account the meticulous comments and suggestions of the reviewers. I sincerely appreciate your attentive efforts to welcome and respond to our criticisms, some strict but constructive, and to implement our suggestions that would make the definitions, concepts, content and expressions of the various parts as well as discussion section of the article clearer and more understandable. Of course, my greatest gratitude and appreciation belongs to the reviewers who generously offered their valuable time, criticism and suggestions for the improvement of this article, with the sole motivation of contributing to science and humanity.
---

## [Editor Report · Acceptance letter]

2 Jul 2024

PONE-D-23-34799R1 

PLOS ONE

Dear Dr. Sauvé, 

I'm pleased to inform you that your manuscript has been deemed suitable for publication in PLOS ONE. Congratulations! Your manuscript is now being handed over to our production team.

Kind regards, 

on behalf of

Prof. Dr. Sukru Torun 

Academic Editor

PLOS ONE